# Self-assembly of sustainable plant protein protofilaments into a hydrogel for ultra-low friction across length scales
Olivia Pabois[1,7], Yihui Dong[2,7], Nir Kampf [2], Christian D. Lorenz [3], James Doutch[4], Alejandro Avila-Sierra[5], Marco Ramaioli[5], Mingduo Mu[1], Yasmin Message[1], Evangelos Liamas[1,6], Arwen I. I. Tyler [1], Jacob Klein [2] & Anwesha Sarkar [1] ✉

Designing plant protein-based aqueous lubricants can be of great potential to achieve sustainability objectives by capitalising on inherent functional groups without using synthetic chemicals; however, such a concept remains in its infancy. Here, we engineer a class of self-assembled sustainable materials by using plant-based protofilaments and their assembly within a biopolymeric hydrogel giving rise to a distinct patchy architecture. By leveraging physical interactions, this material offers superlubricity with friction coefficients of 0.004-to-0.00007 achieved under moderate-to-high ($10^2$-to-$10^3$ kPa) contact pressures. Multiscale experimental measurements combined with molecular dynamics simulations reveal an intriguing synergistic mechanism behind such ultra-low friction - where the uncoated areas of the protofilaments glue to the surface by hydrophobic interactions, whilst the hydrogel offers the hydration lubrication. The current approach establishes a robust platform towards unlocking an untapped potential of using plant protein-based building blocks across diverse applications where achieving superlubricity and environmental sustainability are key performance indicators.

Achieving superlubricity or near-zero friction with ultra-low sliding friction coefficients (<0.01) is a revolutionary engineering paradigm for energy saving[1,2] and biomedical applications[3]. Developing functional materials by exploiting the untapped potential of hydration lubrication achieving oil-free superlubricity at moderate-to-high contact pressures mimicking those found in biology (such as mucinous self-assemblies in healthy oral epithelial surfaces[4] or synovial fluids in articulating cartilage surfaces in healthy human joints[5]) seems to be an obvious alternative for achieving a sustainable future. Although literature on hydration lubrication has surfaced in the past decade, engineering of eco-friendly, efficient, functional aqueous lubricants remains far from realisation. Many, if not most, aqueous lubricants showing superlubricity employ unilamellar vesicles[6], solid lubricants[7], poly-zwitterionic brushes[8], or amphiphilic surfactants[9] that are nearly exclusively derived from synthetic chemistry. Although the recent use of hydrogels offers a unique route for achieving ultra-low friction whilst promoting industrial sustainability objectives, often these hydrogels are prepared

following synthetic routes and using lipids, polyacrylamides, and other zwitterionic synthetic polymers[10–14]. Therefore, seeking an environmentally friendly source of advanced functional lubricants showing superlubricity remains a critical bottleneck.

Using plant-derived proteins as alternative building blocks to synthetic aqueous lubricants could be an ideal way to address this sustainability issue as they can be naturally sourced in abundance with lower carbon footprints[15]. Although plant proteins suffer from functionality issues owing to their complex quaternary structure and limited solubility, there has been an increased momentum to convert them into functional materials such as microgels, nanostructured films, and amyloid fibrils (fabricated by us[16] and other groups[17,18]) exploiting hydrophobic and coulombic interactions manipulating the physical ordering of their naturally occurring functional amino acids. Nevertheless, converting these sustainable plant proteins into sustainable, functional aqueous lubricant materials achieving superlubricity remains to be demonstrated.

[1]School of Food Science and Nutrition, University of Leeds, Leeds, LS2 9JT, UK. [2]Department of Molecular Chemistry and Materials Science, Weizmann Institute of Science, 76100 Rehovot, Israel. [3]Department of Engineering, King's College London, London, WC2R 2LS, UK. [4]ISIS Neutron and Muon Source, Science and Technology Facilities Council, Rutherford Appleton Laboratory, Didcot, OX11 ODE, UK. [5]Université Paris-Saclay, INRAE, AgroParisTech, UMR SayFood, 91120 Palaiseau, France. [6]Unilever Research & Development Port Sunlight, Quarry Road East, Bebington, Merseyside, CH63 3JW, UK. [7]These authors contributed equally: Olivia Pabois, Yihui Dong. ✉e-mail: a.sarkar@leeds.ac.uk

In light of these considerations, the present study reports the first design of a self-assembly of plant protein-derived protofilaments within a biopolymeric hydrogel network as an advanced functional aqueous lubricant material offering superlubricity across length scales. Particularly, we use potato protein, which is a highly biocompatible, non-allergenic plant protein derived from the starch industry as a by-product[19], to create such protofilaments using a highly facile physical crosslinking method, and then electrostatically assemble them with highly hydrating biopolymeric hydrogels. Striking results from this study demonstrate that a robust liquid-vanishing friction with friction coefficients of ca. $4 \times 10^{-3}$ to $7 \times 10^{-5}$) can be achieved under moderate-to-high contact pressures of 300 kPa to 3 MPa due to the synergistic action of the protofilaments offering the surface-anchoring properties, and the hydrogel providing the water mesh-induced hydration, thus behaving like a polymer brush, but made using ecofriendly materials. Such outstanding lubrication properties are due to the 'patchy architecture' of this highly viscoelastic and extensible self-assembled structure, where the areas of the protofilaments uncoated by the hydrogel are available to glue to the surface.

The molecular structure of the bulk and interfacial tribofilms formed by this unique self-assembly were characterised across different length scales combining a wide array of techniques, including imaging (transmission electron (TEM) and atomic force (AFM) microscopy), and scattering (light and small-angle neutron scattering (SANS)). Molecular dynamics (MD) simulations were performed both in the bulk phase and at the surface to gain further insights into the structural mechanism governing the interaction between both components and with the surface. We also thoroughly assessed the lubricant friction-reducing capability at multiscale, using both mini traction machine (macroscale study) and surface force balance (SFB) (nanoscale study). Complementary experiments aiming to get further insight into the adsorption behaviour and viscoelasticity (upon shear and extensional deformation) of the engineered self-assembled structures were additionally conducted with a quartz-crystal microbalance with dissipation monitoring (QCM-D), as well as with rotational and extensional rheometers, in order to subsequently define the key parameters influencing hydration performance. This biocompatible ecofriendly aqueous lubricant showing superlubricity at multiple length scales is a key milestone towards creating highly sustainable, plant-based aqueous lubricant materials in the energy sector[20], as well as the next generation of engineered biomedical materials, such as artificial synovial fluid, tear and saliva for lifetime lubrication of natural biological contacts.

## Results and discussion

### Molecular structure of the self-assembled protofilament-hydrogel

To comprehensively explore the protofilament-hydrogel assembling mechanism, the molecular architecture of each individual component (i.e., the potato protein protofilament (PoPF) and the hydrogel made using naturally occurring biopolymers (i.e., xanthan gum (XGH) or $\kappa$-carrageenan

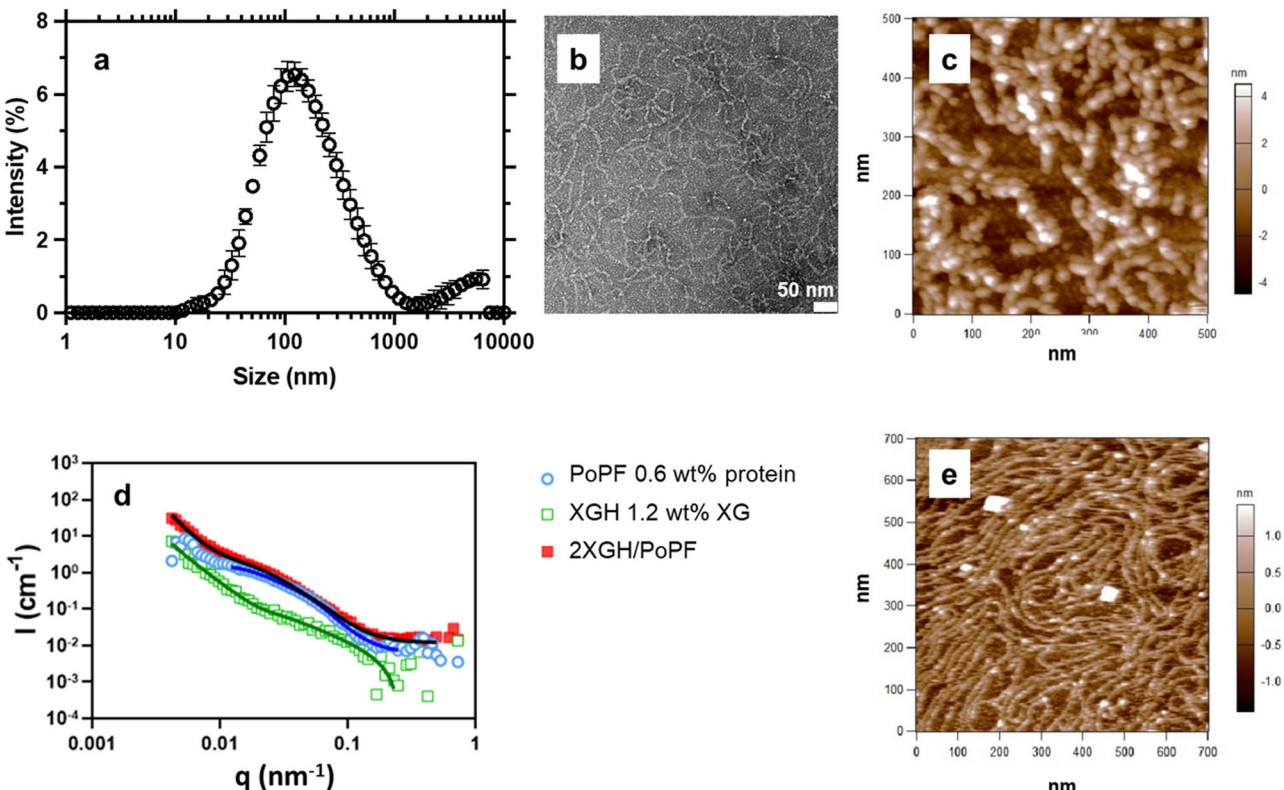

**Fig. 1 | Molecular structure of the self-assembled protofilament-hydrogel.**
**a** Particle size distribution of potato protein protofilament (PoPF) obtained from dynamic light scattering (DLS) measurements. PoPF displays a hydrodynamic diameter of ca. $d_H = 100$ nm and a surface charge of $\zeta = +30$ mV. PoPF nanostructure observed by (**b**) transmission electron microscopy (TEM), following negative staining, and (**c**) atomic force microscopy (AFM), following sample deposition onto a negatively charged, hydrophilic (mica) surface and immersion into citrate buffer (pH 3.0). The TEM scale bar is 50 nm. Patatin (main potato protein unit[21]) monomers seem to aggregate with each other, ultimately forming a nanofibril-like protofilament structure. **d** Scattered intensity ($I$) as a function of the scattering vector ($q$) for the potato protein protofilament/xanthan gum hydrogel (2 XGH/PoPF) and each individual component (1.2 wt% XGH, and 0.6 wt% PoPF), at 25 °C, measured by small-angle neutron scattering (SANS). Solid lines correspond to fits to the data obtained using the unified model[29,30]. **e** Nanostructure of the self-assembled protofilament/-hydrogel (2 XGH/PoPF) observed by AFM, following sample deposition onto a negatively charged, hydrophilic (mica) surface and immersion into citrate buffer (pH 3.0). A continuously connected protofilament-containing hydrogel network seems to form. Both XGH and the mica surface being negative charge, no adsorption was expected, therefore preventing any structure from being visible once measuring XGH on its own (Supplementary Fig. 1). Each measurement was reproduced at least three times; the average measurement is shown.

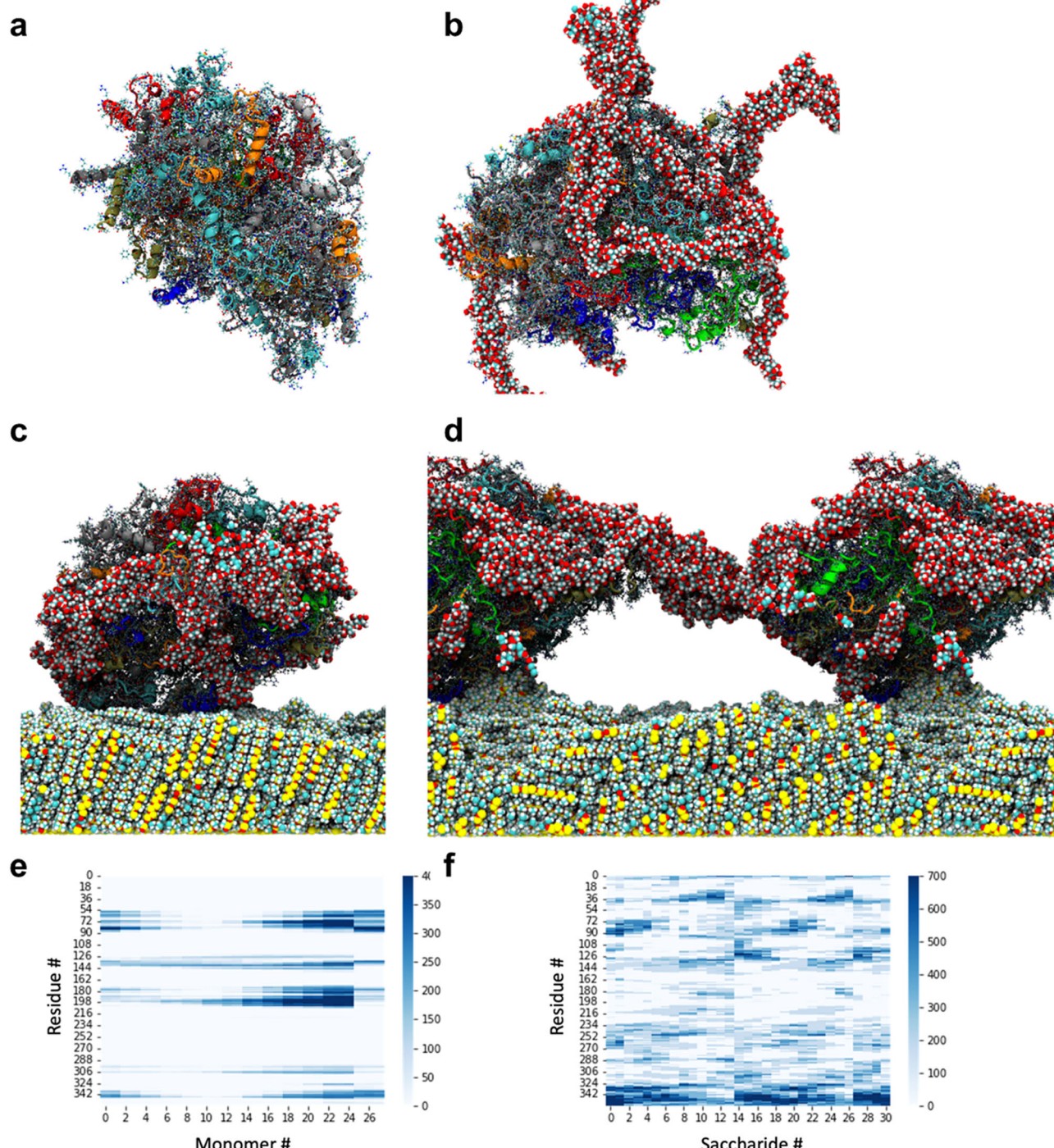

**Fig. 2 | Molecular dynamics (MD) simulations of the interaction between PoPF and XGH, in an aqueous dispersion and in contact with a polydimethylsiloxane (PDMS) surface. a** Snapshot of the MD simulation of a single potato protein (patatin)-based protofilament (PoPF) in an aqueous solution (radius of gyration ($R_g$) = 46.1 ± 0.1 Å, eccentricity = 0.06 ± 0.01). **b** Snapshot of the MD simulation of xanthan gum (XG) partially coating PoPF in an aqueous solution ($R_g$ = 44.8 ± 0.1 Å, eccentricity = 0.15 ± 0.01). The XG molecules are represented by the chains of spheres, which are coloured by their element: cyan (carbon), red (oxygen) and white (hydrogen). PoPF is represented by a cartoon that shows the secondary structure of the potato protein (patatin). The cartoon of each patatin in PoPF is coloured differently. **c, d** Snapshots from the MD simulation of PoPF interacting with XGH in the presence of a PDMS surface, where two PoPF are connected by XGH, whereas the naked (i.e., uncoated by XGH) part of PoPF interacts with PDMS. The PDMS is in the form of a slab, and its molecules are shown as spheres where cyan (carbon), silicon (yellow), red (oxygen) and white (hydrogen) are used to denote the different atoms in the polymer. Contact maps for (**e**) PoPF and PDMS and (**f**) PoPF and XGH.

(KCH) hydrogels) and their combination (i.e., 2 XGH/PoPF and 2 KCH/PoPF) was resolved at both the nano- and atomistic scales, by employing a very wide range of structural techniques, including dynamic light scattering (DLS) (Fig. 1a), transmission electron microscopy (TEM) (Fig. 1b), atomic force microscopy (AFM) (Fig. 1c, e and Supplementary Fig. 1), small-angle

neutron scattering (SANS) (Fig. 1d), and molecular dynamics (MD) simulations (Fig. 2 and Supplementary Figs. 2–4). To this end, we designed a comprehensive protofilament fabrication process using potato protein and induced their self-assembly within a polysaccharide hydrogel (either XGH or KCH)) (see 'Experimental' Section). The characterisation of both the bulk

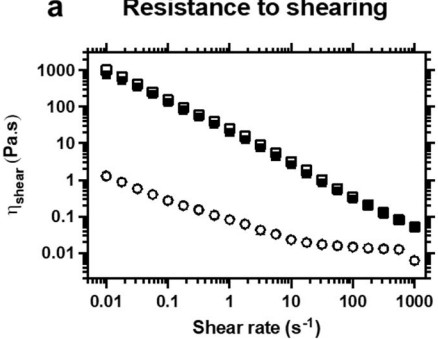
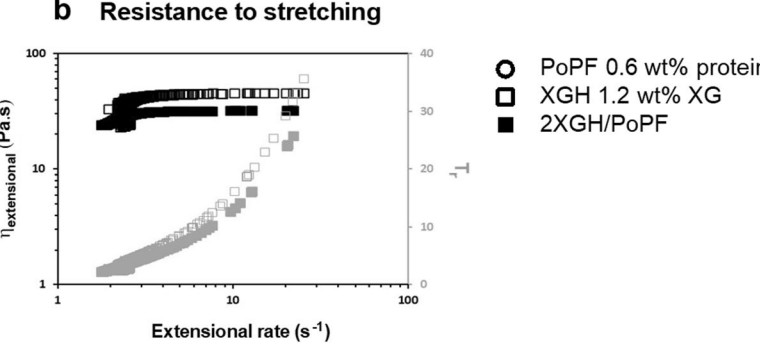

**Fig. 3 | Viscoelastic behaviour of the self-assembled protofilament/hydrogel.**
Evolution of (**a**) the shear viscosity ($\eta_{shear}$) as a function of the shear rate, obtained from stress-controlled rotational rheometry measurements, and (**b**) the apparent extensional viscosity ($\eta_{extensional}$) and Trouton ratio ($T_r$) as a function of the extensional rate, obtained from extensional rheometry measurements, performed on the potato protein protofilament/xanthan gum hydrogel (2 XGH/PoPF) and each individual component (1.2 wt% XGH, and 0.6 wt% PoPF), at 37 °C. The self-assembled protofilament/hydrogel exhibits a high-viscosity, shear-thinning behaviour, very similar to that of XGH on its own; in comparison, PoPF displays much lower shear viscosity values. While PoPF was found to break at an extremely short time (ca. $t_b$ = 0.014 s), thereby preventing the assessment of its stretchability, both the self-assembled protofilament/hydrogel (2 XGH/PoPF) and XGH show a rise in both the apparent extensional viscosity and $T_r$ upon increasing extensional rates, more notable so for the latter; for both, particularly the self-assembled system, elastic forces predominate over viscous ones (as $T_r$ > 3). Each experiment was reproduced at least three times; the average and a representative measurement are shown for shear and extensional rheology, respectively.

(DLS, SANS, and MD simulations) and interfacial film (TEM, AFM, and MD simulations) structures formed by the protofilament-hydrogel systems ultimately enabled shedding light on the molecular process governing their superlubricity. Whilst we mainly focused on using XGH as the hydrogel component, we also outline the importance of the polymer type by pin-pointing the specific differences observed using another polysaccharide hydrogel (KCH) where appropriate (Supplementary Fig. 1).

DLS measurements reveal that PoPF exhibits a monomodal size distribution (Fig. 1a). Both TEM (Fig. 1b) and AFM (Fig. 1c and Supplementary Fig. 1) images reveal that the 65 °C heat treatment of the 6 wt% potato protein isolate solution fabricated at pH 3.0 results in the formation of long, curly nanofibrillar structures via the aggregation of sphere-shaped patatin (main potato protein unit[21]) particles, with diameter and height sizes of ca. 20 nm and 3–5 nm, respectively.

Both images have the same scale length (ca. 500 nm), but the sphere-shaped patatin particles are about 10 times larger in the AFM micrograph compared to TEM. The fibrillar length could not be determined given the high level of entanglement. The topographical image (Fig. 1c) is consistent with observations made with a range of globular, plant-based proteins that are physically hydrolysed at low pH and high temperature[19,22–27], including potato protein for which similar morphologies and dimensions were obtained[19]. Under these conditions, fibrillisation is well known to occur following a 'polypeptide model', whereby proteins first hydrolyse into shorter, lower molecular weight peptide fragments, which subsequently undergo intermolecular interactions due to denaturation-induced unfolding, and oligomerise into protofilaments, ultimately assembling into amyloid fibrils[17,19,22–27].

Herein, the fibrillar structure formed by PoPF seems to comprise only one protofilament (i.e., no assembled and twisted morphology), therefore not reaching the amyloid-like fibril stage (hence the naming protofilaments). XGH unlike PoPF shows no prominent topographical features (Supplementary Fig. 1). This was expected as XGH is negatively charged ($\zeta$ = −40 mV[28] and thereby cannot adhere to the negatively charged mica surface, contrary to the positively charged PoPF ($\zeta$ = +30 mV). SANS data analysis using the unified model[29,30] (Fig. 1d) reveals that PoPF exhibits a radius of gyration ($R_g$) of 75.4 ± 1.5 Å with a fractal dimension of ca 3.3, thus suggesting the formation of structures with a quite rough surface—in agreement with observations made through microscopy (Fig. 1b, c), while with its power of 1.1, XGH seems to be composed of rod-like objects with a $R_g$ of 156.4 ± 4.1 Å forming a quite branched, mass fractal network. Although PoPF seems to form fibril-like structures, the system could not be fitted to a cylinder-type model very likely because of its too high polydispersity and branching. Surprisingly, the combination of PoPF and XGH leads to the formation of convoluted, continuous interconnected filaments that seem to cover the mica surface (Fig. 1e and Supplementary Fig. 1). This suggests that PoPF is most likely coated in a 'patchy way' by XGH, and catalyses the surface binding of this self-assembly by virtue of its cationic surface charge, whereas XGH glues one protofilament to another, ultimately evolving into a dense, electrospun fibre-like, highly ordered filamentous network. The analysis of data corresponding to their combination suggests objects with a rough surface ($R_g$ = 74.2 ± 2.5 Å, and fractal dimension of ca. 2.9), which may self-assemble in the shape of large aggregates (Fig. 1d). Noteworthy, this long, stranded architecture occurring due to XG rigid rods is a signature of 2 XGH/PoPF system; instead, the other polysaccharide (KC) seems to form fractal aggregates when mixed with PoPF (Supplementary Fig. 1).

## MD simulations both in the bulk and at the interface

To further support the proposed patchy model of PoPF coated by XGH, MD simulations were carried out both in the bulk phase and at the interface, to gain an atomistic description of the interactions taking place between PoPF and the XGH aqueous medium, as well as with a highly hydrophobic, polydimethylsiloxane (PDMS) surface. To the best of our knowledge, this is the first MD simulation study performed on a self-assembled system made up of protofilaments and hydrogels. Using 8 patatin molecules to model a protofilament (Fig. 2a, see 'Experimental' section), the snapshots obtained from the simulations clearly show that the XG molecules coat only parts of the PoPF moieties, with the patatin in the PoPF moiety interacting with the XG molecules via its C-terminal end (Fig. 2b). Remarkably, PoPF is adsorbed onto the surface of the PDMS substrates via its 'naked', uncoated area (Fig. 2c) and XG molecules are able to form bridges between neighbouring PoPF entities (Fig. 2d). We then identified any pair of residues (which are defined as a monomer in PDMS, an amino acid in the potato protein, and a saccharide in XG) in the different molecules to be in contact if their centre of mass was less than 1.5 nm away from a neighbouring molecule.

The contact map in Fig. 2e shows that PoPF binds to the PDMS surface specifically through residues 72 and 90, and through residues 180 and 200. Both of these regions of the protein contain high densities of hydrophobic residues which may shield themselves from the aqueous environment by interacting with the polymeric interface. The interaction of PoPF with the PDMS substrate causes the PoPF moieties to deform and become both more spherical (eccentricity of 0.13 ± 0.01 at the PDMS surface vs. 0.15 ± 0.01 in solution) and more compact ($R_g$ of 43.6 ± 0.1 Å at the PDMS surface vs.

**Fig. 4 | Macroscale lubrication performance and adsorption behaviour of the self-assembled protofilament/hydrogel. a** Speed-dependent evolution of the friction coefficient, obtained from tribology measurements performed with non-charged, hydrophobic (PDMS) surfaces (at the macroscale), on the self-assembled potato protein protofilament/xanthan gum hydrogel (2 XGH/PoPF) and each individual component (1.2 wt% XGH, and 0.6 wt% PoPF), at 37 °C. The lubrication properties of citrate buffer are also shown for comparison purposes. 2 XGH/PoPF shows an outstanding lubrication performance both in the boundary and hydrodynamic regions, exhibiting ultra-low friction coefficients contrary to both PoPF and XGH on their own. **b** Influence of the pH on the lubrication performance of the self-assembly system (2 XGH/PoPF) assessed by macroscale tribology. pH was changed from 3.0 to 5.0, upon addition of 1.0 M NaOH, and was increased from pH 3.0 to 7.0, before being decreased back to 3.0 with 1.0 M HCl. The superlubricity of 2 XGH/PoPF was retained at a pH below neutral pH (i.e., below the potato protein isolate isoelectric point (pI of ca. 5)). **c** Time-dependent evolution of the resonance frequency ($\Delta f$) measured using QCM-D, upon adsorption of the self-assembled potato protein protofilament/xanthan gum hydrogel (2 XGH/PoPF) and each individual component (1.2 wt% XGH, and 0.6 wt% PoPF), in the presence of a PDMS-coated surface. Each sample was injected into the chamber, which was then rinsed with buffer to assess the sample ability to remain adsorbed to the surface following buffer rinsing. For readability purposes, resonance frequencies are only shown for the 5th overtone. Each sample was diluted 20 times prior to any measurements. Both the self-assembled protofilament/hydrogel system and its individual components readily adsorb onto the PDMS surface, with 2 XGH/PoPF showing a much higher extent of adsorption. Each measurement was reproduced at least three times; the average measurement is shown.

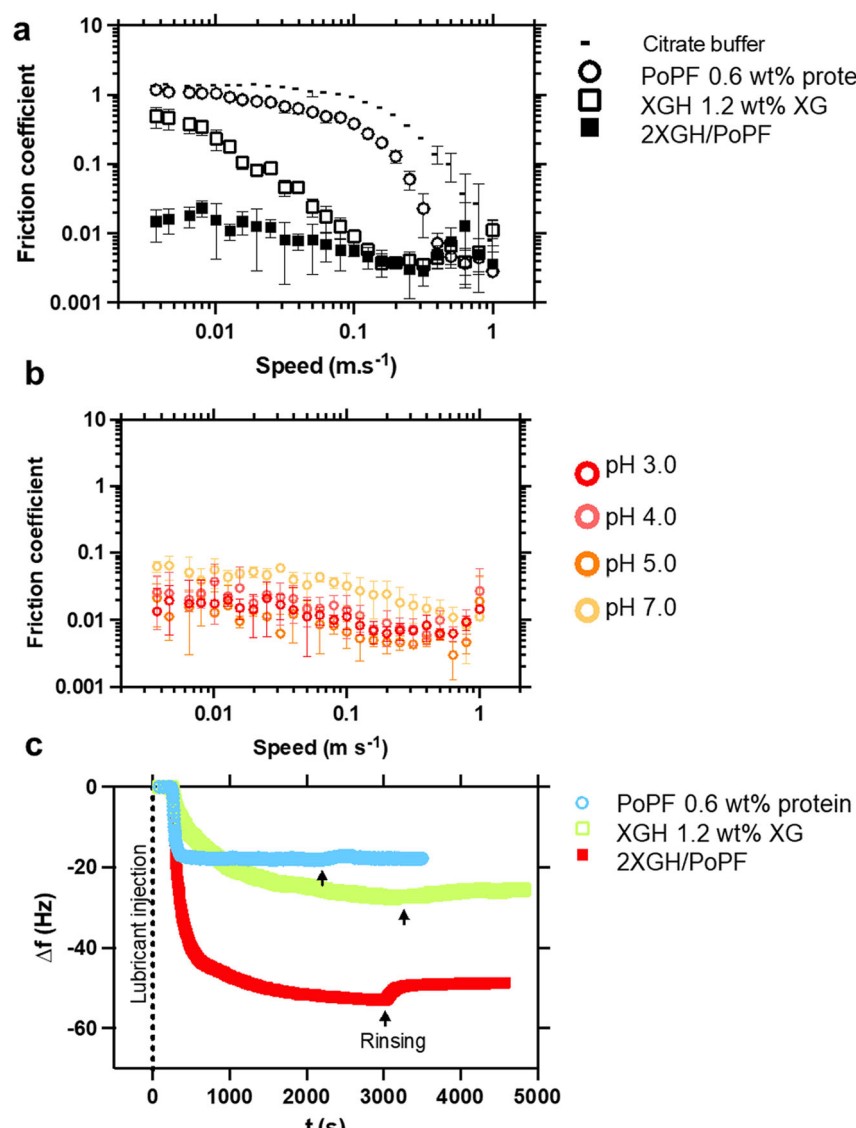

44.8 ± 0.1 Å in solution). Also, the interactions between the patatins in PoPF and XGH (Fig. 2f) change, as residues 72–90 in the patatins, which had been interacting with XG in the aqueous phase (primarily through the positively charged residues ARG 70 and LYS 76), preferentially interact with the PDMS polymers (Fig. 2e). When PoPF is adsorbed to the PDMS substrate, XG seems to bind primarily to the patatins through interactions with the C-terminal end (residues 327–359) of the potato protein, which has a particularly high density of positively charged residues (LYS 327, LYS 328, LYS 343, ARG 344, LYS 347, ARG 352, LYS 353, LYS 354, ARG 356, and LYS 359). When XG is bound to PoPF before it adsorbs to the polymer interface, we also observe interactions between residues 113–137 (there are three LYS residues and several polar residues within this region) and the XG molecules which are not available as a result of PoPF adsorption to the PDMS interface. The patatins are generally found to interact via the same regions, which have a high density of hydrophobic residues interacting with the PDMS surface, whether they are in an isolated protofilament (Supplementary Fig. 2a), interacting with xanthan gum (Supplementary Fig. 2b), or interacting with both XGH and PDMS (Supplementary Fig. 2c). Additionally, XG molecules were shown to interact in a head-to-tail arrangement that facilitates fibre-like formations, such that the one end of a given XG molecule interacts with the opposite end of another molecule (Supplementary Fig. 3). In summary, the MD simulations (Fig. 2) provide insight into the molecular mechanisms that drive the patchy architecture of XGH

on PoPF, which was also observed in the topographical images. This patchy architecture seems to be primarily driven by significant binding to the domains of the proteins that have a high density of cationic residues, while the binding of PoPF to the PDMS surface is as expected driven by hydrophobic interactions between domains of the protein with high densities of hydrophobic residues. It is now worth asking whether such a unique patchy architecture of the self-assembled structure offers synergistic lubricity benefits; this will be discussed later.

## Rheological performance

Macromolecular hydrodynamic effects have been considered as a key contributor to lubrication mechanism in biological contact surfaces[4,31–33]. In order to understand the role of viscous lubrication in the overall lubricity performance, the resistance to shearing and stretching of the self-assembled protofilament/hydrogel was measured using rotational (Fig. 3a and Supplementary Fig. 5a) and extensional (Fig. 3b and Supplementary Fig. 5b) rheometry, respectively.

The shear viscosity evolution was monitored over a broad range of shear rates (0.01–1000 s$^{-1}$) (Fig. 3a). Irrespective of the type of polymer (XG or KC) used to prepare the hydrogel (Fig. 3a and Supplementary Fig. 5a), both the self-assembled system (2 XGH/PoPF or 2 KCH/PoPF) and the individual components (PoPF, XGH, and KCH) display a shear-induced viscosity decrease over the whole range of shear rates studied, thereby

exhibiting a shear-thinning behaviour. Nonetheless, compared to PoPF, whose shear viscosity does not exceed $\eta_{shear} = 1.3 \pm 0.1$ Pa.s at $0.01$ s$^{-1}$, and diminishes of only two orders of magnitude upon reaching $1000$ s$^{-1}$, stabilising at a near-plateau from ca. $10$ s$^{-1}$, XGH and 2 XGH/PoPF were found to display a strikingly sharper (four orders of magnitude), and similar, shear rate-dependent decrease (from $\eta_{shear} = 1006 \pm 122$ Pa.s for XGH, and $757 \pm 23$ Pa.s for 2 XGH/PoPF, at $0.01$ s$^{-1}$, to $0.052 \pm 0.003$ Pa.s for XGH, and $0.050 \pm 0.002$ Pa.s for 2 XGH/PoPF, at $1000$ s$^{-1}$). Compared to 2 XGH/PoPF (Fig. 3a), 2 KCH/PoPF displays an order of magnitude lower viscosity, particularly at $0.01$ s$^{-1}$ (Supplementary Fig. 5a); such an observation might be associated with the structural differences of the continuous, compact filaments offering more viscosity in contrast to the fractal aggregates (Supplementary Fig. 1). Of more importance, the viscosity of the self-assembled protofilament/hydrogel overlaps with that of the hydrogel alone, independently of the polymer types or shear rates tested (Fig. 3a and Supplementary Fig. 5a). This result highlights that the lubricity difference (if any) between the individual components and the self-assembled system might be predominantly linked to the surface behaviour rather than the viscous behaviour, the latter, being largely dominated by the hydrogel component.

Changes in capillary thread shape (Supplementary Fig. 6) and diameter (Supplementary Fig. 7) upon extensional deformation were recorded over time, and the evolution of the apparent extensional viscosity ($\eta_{extensional}$) and $T_r$, which are characteristics of the lubricant viscoelasticity) (Fig. 3b and Supplementary Fig. 5b) were plotted against extensional rates.

While the extensional rheology of PoPF could not be assessed due to its too short breaking time (ca. $t_b = 0.014$ s), XGH and the 2 XGH/PoPF systems were found to form very long-lived and slender filaments (Supplementary Figs. 6 and 7). Contrary to rotational rheology measurements, which show similar shear viscosity evolutions upon increasing shear rates for the XGH-based samples, extensional rheology measurements demonstrate better resistance to thinning ($t_b = 0.67 \pm 0.37$ s for 2 XGH/PoPF vs. $t_b = 0.51 \pm 0.22$ s for XGH), lower extensional viscosity values (ca. $\eta_{extensional} = 24.0 \pm 9.1$ Pa.s for 2 XGH/PoPF vs. ca. $\eta_{extensional} = 44.2 \pm 27.3$ Pa.s for XGH), and lower Trouton ratio values (ca. $T_r = 21.0 \pm 2.8$ for 2 XGH/PoPF vs. ca. $T_r = 30.6 \pm 6.8$ for XGH) for the self-assembled system (Fig. 3b). Similar trends, but lower in magnitude, were observed in terms of both shear and extensional rheology when changing the polysaccharide type from XGH to KCH (Supplementary Fig. 5).

## Superlubricity

Usually, friction force measurements are carried out at only one scale, even though lubricity and the value of friction coefficients are known to be highly scale dependent[34]. In order to paint a full picture on the lubrication property of these new self-assembled protofilament/hydrogel systems, friction forces were measured at the macroscale using a PDMS ball-on-disk tribometer, with a contact pressure of 300 kPa[35], which is representative of lower bound pressures found in biological conditions (Fig. 4 and Supplementary Fig. 9), and surface force balance (SFB) measurements were conducted at the nanoscale using mica surfaces, with much higher contact pressures (ca. 3 MPa[36]), which are more representative of articular joint conditions (Fig. 5 and Supplementary Figs. 11 and 12) (see 'Experimental' section).

Unlike plant protein-based microgels[16], PoPF displays high friction coefficient values over the whole entrainment speed range, reaching up to $1.18 \pm 0.05$ at $0.004$ m s$^{-1}$, similarly to citrate buffer at pH 3.0; instead, XGH seems to induce better lubrication, particularly in the hydrodynamic regime, where friction coefficients as low as $0.009 \pm 0.002$ are obtained at $0.1$ m s$^{-1}$ (Fig. 4a). Surprisingly, the self-assembly of the protofilaments with the hydrogel synergistically results in an unusual, nearly speed-independent friction curve, with friction coefficient values as low as $0.004 \pm 0.005$ in the boundary regime (at $0.004$ m s$^{-1}$) (Fig. 4a). When plotted against minimum hydrodynamic film thickness ($h_{min}$) (Supplementary Fig. 8), the friction curves of the constituents overlapped with that of the buffer. On the

contrary, the self-assembly (2 XGH/PoPF system) showed ultra-low friction independent of the film thickness, showing effective boundary films capable of maintaining low friction forces at low surface separations. Of more interest, pH does not seem to affect the lubrication properties of this self-assembled system, an increase in friction being observed only at a neutral pH of 7.0, where both PoPF and XGH are negatively charged and thus unable to interact with each other (Fig. 4b). Such an ultra-low macroscopic lubrication performance is also observed when decreasing the polysaccharide/protein ratio from 2 to 0.5 (Supplementary Fig. 9a) and changing the polysaccharide type from XGH to KCH (Fig. S9b). Based upon the complementary interfacial data obtained via QCM-D measurements with PDMS-coated silicon surfaces, the remarkably better lubrication properties exhibited by the self-assembled protofilament/hydrogel may be explained by its 3 times stronger ability to adsorb onto the surface and to remain attached following rinsing, compared to that of the polysaccharide hydrogel and protein protofilaments on their own (Fig. 4c). Such a high adsorption capacity is rather specific to the use of XGH rather than KCH (Supplementary Fig. 10), thus highlighting the importance of the filamentous structure formed in the self-assembled 2 XGH/PoPF system (Fig. 1e) and of its patchiness visualised through MD simulations (Fig. 2). Such an efficient friction-reducing mechanism can be attributed to the synergy between: PoPF almost behaving like a colloidal glue attaching to the surface, and XGH network sporadically interacting with the protofilaments forming a water-encapsulating macromolecular brush, ultimately providing hydration. To further investigate the hydration lubrication provided by this system, and clarify whether such ultra-low friction values were maintained in a high contact pressure situation, SFB measurements were conducted with the 2 XGH/PoPF system.

The normal ($F_n(D)$) and shear ($F_s$) force profiles, as well as the absolute surface separation between two mica surfaces ($D$) across three different systems: PoPF, XGH, and their self-assembly (2 XGH/PoPF), were measured with SFB. The normal force ($F_n$) vs. surface separation ($D$) profiles between the mica surfaces bearing PoPF, XGH, and the self-assembled 2 XGH/PoPF system, across each dispersion in citrate buffer were recorded by SFB (see 'Experimental' section)[37,38], and plotted in Fig. 5 as $F_n(D)/R$ in accordance with the Derjaguin approximation[37,38], to normalise for slightly different radii of curvature ($R$) of the mica sheets.

In the case of PoPF alone on both surfaces (Fig. 5a), a monotonic long-ranged repulsion commencing at ca. $D = 600$ nm is observed and may be attributed to steric interactions due to loose multilayer PoPF adsorbed on the surface, as indicated by the AFM micrographs (Fig. 1c) and the broad DLS size distribution (Fig. 1a). A sharper rise in $F_n$ starts at $D <$ ca. 300 nm, likely due to the compaction of the initially looser adsorbed PoPF. At the strongest compressions attained in our measurements, the surfaces reach a 'hard-wall' separation, $D_{HW} = 300 \pm 1$ nm for both the first (open symbols) and subsequent (filled symbols) approach. This is equivalent to ca. 150 nm of compacted PoPF on each surface, likely in the form of multiple compacted protofilaments of broad size distribution, as suggested by the broad DLS distribution (Fig. 1a) and AFM micrographs (Fig. 1c).

In the case of XGH alone on both surfaces (Fig. 5b), both the magnitude of the exponentially decaying, long-ranged repulsion, as well as the range of the strong steric repulsion, were much shorter compared to PoPF, with steric forces commencing at separations $D \leq 50$ nm. This suggests a compact and/or very weakly adsorbed layer of XGH on each surface. This is expected as the negatively charged XGH[28] would not adsorb to the negatively charged mica surfaces, but weak adsorption due to short-ranged van der Waals attraction at high buffer salt concentration may be possible. The surface separation ($D_{HW}$) under progressively high compression is then reduced from ca. 25 nm to 0 nm. This indicates that XGH is fully squeezed out from the two mica surfaces under compression, in line with the expected very weak (if any) adsorption between XGH and the similarly (negatively) charged mica surfaces. Such a behaviour resembles the macroscale behaviour observed with high friction coefficient values (Fig. 4a) supported by the limited adsorption observation with both AFM (Supplementary Fig. 1) in the presence of mica surfaces and QCM-D (Fig. 4c) in the presence of PDMS surfaces.

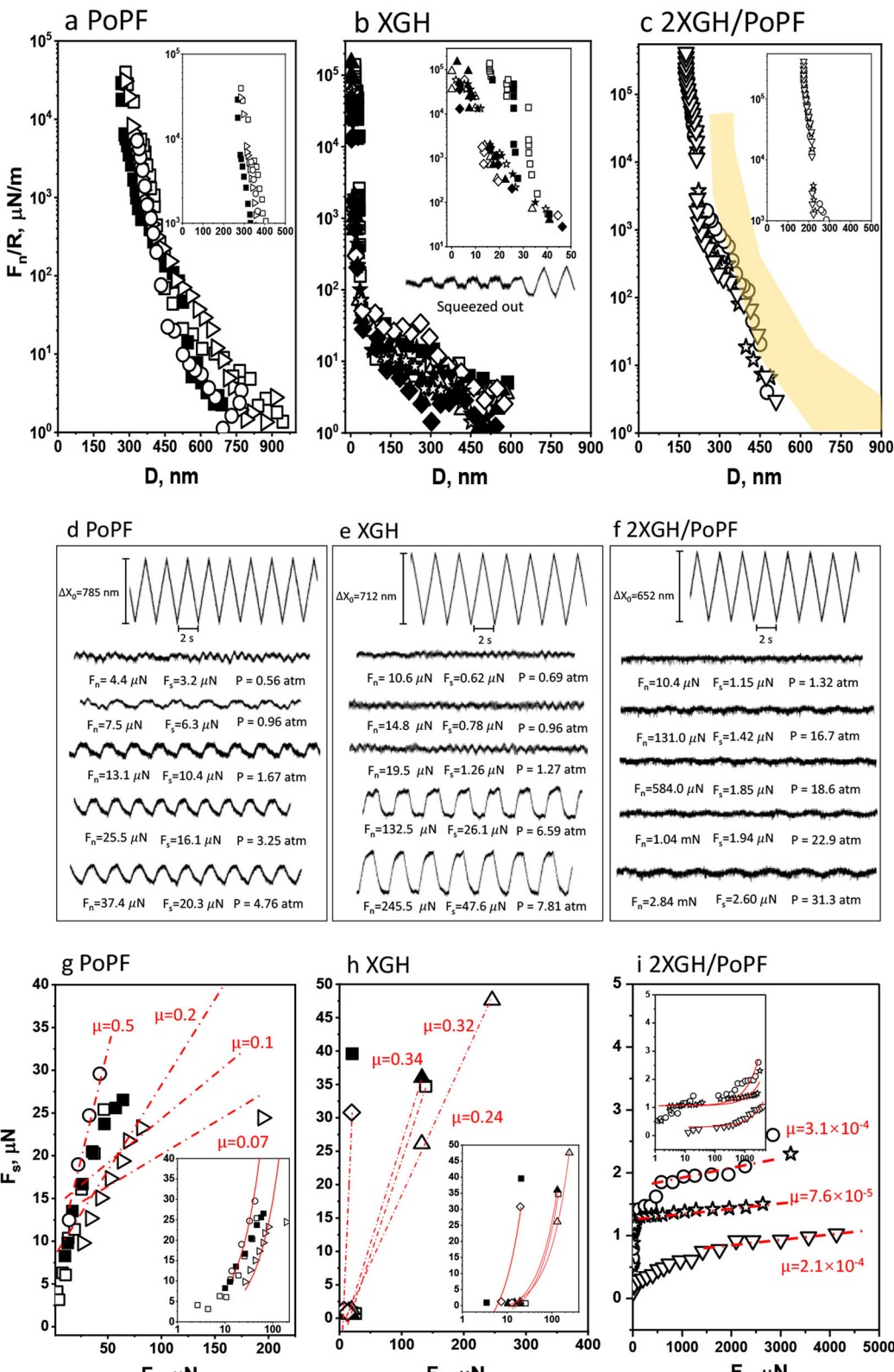

In the case of the self-assembled 2 XGH/PoPF system (Fig. 5c), monotonically repulsive interaction commences at ca. 500 nm, similarly to PoPF alone (Fig. 5a, see the shaded region in Fig. 5c). This likely occurs for similar reasons, i.e., large PoPF adsorbed on the mica surface, though now complexed with XGH. It is of interest that the surface separation due to steric repulsion at the highest applied loads is significantly lower for the 2

XGH/PoPF combination (ca. $D_{HW}$ = 200 nm) than for PoPF alone (for which ca. $D_{HW}$ = 300 nm). This indicates that the complexation with XGH may result in lower PoPF adsorption on the negatively charged mica, due to the negative charge on XGH, as well as possibly due to changes in the aggregate structures. Importantly, no squeezing out was seen up to the highest loads applied—which correspond roughly to ca. 30 atm (i.e., 3 MPa)

**Fig. 5 | Nanoscale lubrication performance of the self-assembled protofilament/hydrogel.** Normal force ($F_n$), normalised by the radius of curvature ($R$ = ca. 1 cm), plotted against the surface separation distance between curved mica surfaces ($D$), interacting across: **a** potato protein protofilaments (0.6 wt% PoPF), **b** xanthan gum hydrogel ((0.15 wt% XGH), and **c** the self-assembled protofilament/hydrogel (2XGH/PoPF); the yellow-shaded region summarizes the data from **a**. The insets show the force-distance profiles close to the hard-wall separations, on a magnified scale. Empty and filled symbols correspond to first and subsequent approaches. Typical traces of shear force vs. time for the sliding mica surfaces across: **d** PoPF, **e** XGH, and **f** 2XGH/PoPF. The uppermost trace in each graph shows the back-and-

forth lateral motion applied to the upper surface, whereas the other traces show the forces transmitted to the lower surface at different loads ($F_n$) and friction forces ($F_s$). Each set of traces was recorded during the same approach profile. Shear force ($F_s$) as a function of the normal force ($F_n$) measured between mica surfaces across: **g** PoPF, **h** XGH, and **i** 2XGH/PoPF. Red lines in the main (lin–lin) figure and (lin–log) insets are constant friction coefficients ($\mu = \partial F_s / \partial F_n$). Open and solid symbols represent first and second approaches, respectively. Each measurement was reproduced at least two times, and at least two contact positions were done in each pair of mica sheets; a representative measurement is shown.

of contact pressure between the mica surfaces, suggesting that XGH complexes strongly with PoPF, leaving nonetheless some non-coated areas on PoPF, which in turn adhere strongly to the mica surfaces, as observed in the MD simulations (Fig. 2c, d).

Figure 5d–f illustrates typical shear force ($F_s$) vs. time ($t$) traces at different loads $F_n$ between the mica surfaces across dispersions of PoPF, XGH, and their self-assembly (2 XGH/PoPF) in citrate buffer, in response to the lateral back-and-forth motion applied to the upper surface. We note that typical shear traces across the buffer solution alone show very low frictional dissipation (attributed to hydration lubrication) down to mica-mica contact at high loads (Supplementary Fig. 11), demonstrating that the buffer carrier does not influence the frictional results across the macromolecular dispersions. The shear traces (Fig. 5d–f) reveal clearly the following: frictional forces ($F_s$) increase strongly already at quite low loads ($F_n$) for PoPF (Fig. 5d) and XGH (Fig. 5e), but remain consistently low up to high loads for their self-assembly 2 XGH/PoPF (Fig. 5f), thus showing excellent lubrication. The corresponding variation of $F_s$ vs. $F_n$ is summarised in Fig. 5g–i.

The friction coefficient ($\mu$) values across both (single-component) PoPF (Fig. 5g) and XGH (Fig. 5i) dispersions are on the order of $10^{-1}$, though the origin of the high friction coefficient ($\mu$) values is likely different in the two cases. In PoPF case, the protein is positively charged, implying strong adsorption onto the mica surface (in the form of aggregates, as earlier discussed and seen in the AFM micrograph (Fig. 1c); the poor lubrication very likely results from high-energy dissipation as the PoPF layers slide past each other, suggesting poor hydration of the proteins, which is in line with the macroscale behaviour (Fig. 4a). We believe that it is less likely that bridging of the PoPF aggregates between the mica surfaces occurs, which would also result in high friction, as seen with other bridging macromolecules[39,40]. In the case of the negatively charged XGH (Fig. 5h), molecules may be weakly adsorbed, as noted earlier, to the negatively charged mica surface via Van der Waals interactions effective at the high salt concentration of the buffer carrier, which strongly screens out the electrostatic interactions. In that case the relatively high friction may be attributed to bridging across the inter-surface gap by the long, linear polysaccharide, as observed in earlier studies of adsorbing polymers[39]. At the highest loads, the weakly-adsorbing XGH, also observed in QCM-D studies (Fig. 4b), is largely if not entirely squeezed out, leaving at most a thin trapped layer bridging the gap, with high frictional dissipation as the surfaces slide past each other; or if the XGH is entirely squeezed out (see inset to Fig. 5b), the friction may then be high due to mica-mica contact (Supplementary Fig. 11). The sliding velocity-independence of the friction across XGH solution (Supplementary Fig. 12) also indicates that the high friction observed with XGH alone is likely to be due to a bridging effect rather than a hydrodynamic one, in line with the earlier observed bridging behaviour[39]. The increased friction force with increasing load forces reflects a higher extent of bridging as XGH is squeezed out.

In strong contrast, for the shear across the 2 XGH/PoPF (Fig. 5i), the friction, after an initial sharp rise at the lowest loads, is seen to be very low, reaching friction coefficient ($\mu$) values in the order of $10^{-4}$ to $10^{-5}$ at the highest loads (at which the corresponding mean contact pressures are of ca. 30 atm). We attribute this as follows. Based on the AFM micrographs of the 2 XGH/PoPF system on mica (Fig. 1e), the complexation between XGH and PoPF leads to densely packed, thread-like aggregates of pearl-like units (whose size appears roughly half that of the pearl-like units of the aggregates

of PoPF alone, Fig. 1c). This suggests the following scenario: the 2 XGH/PoPF system attaches strongly to the mica surface thanks to the naked (uncoated) areas of PoPF (for which ζ-potential of +30 mV) via a counterion-release mechanism. These partially surface-attached PoPF parts, however, expose the XGH component at their outer surface, and when these highly hydrated XGH moieties slide past each other, hydration lubrication reduces the frictional dissipation to the extremely low values observed (ca. $\mu = 10^{-4} - 10^{-5}$). Thus, the role of PoPF (via its positive charge) is to anchor the 2 XGH/PoPF aggregates to the negatively charged substrates, while exposing the outer hydrated XGH component allowing significant friction reduction. The initial sharp rise in friction at the lowest loads (Fig. 5i), noted above and seen in similar systems previously[41], is due to viscoelastic dissipation, as the loose PoPF are initially sheared, prior to their compression and compaction at higher loads, on approach and sliding of the surfaces.

In summary, the SFB measurements describing above the interactions between mica surfaces across dispersions of PoPF, XGH, and the 2 XGH/PoPF mixture, together with the AFM micrographs of the corresponding surface structures on mica in these dispersions, shed strong light on the nature of the frictional interactions in these three cases. The high friction conditions are then largely due to molecular interactions (e.g. bond breaking/reforming) as the boundary layers slide past each other, while the low friction arises from much-reduced dissipation due to shear of the sub-nanometre hydration layers in the hydration lubrication mechanism, together with passage over energy barriers as described previously[36] (where phonons are generated leading to the weak dissipation observed). In particular, these studies reveal that the very low friction achieved across the 2 XGH/PoPF system arises from the combined roles of the potato protein protofilaments and the highly hydrated polysaccharide hydrogel: the former acts largely to anchor the surface protofilaments to the (negatively-charged) mica substrate, while the latter provides efficient hydration lubrication by being exposed by the surface-attached protofilaments[42–44]. This underpins and illuminates the molecular mechanisms underlying the findings from the

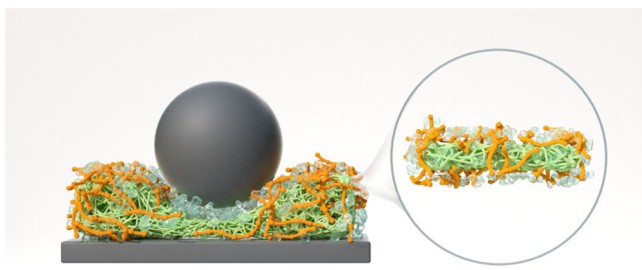

**Fig. 6 | Schematic illustration of hydration lubrication offered by the self-assembly of PoPF and XGH.** The green mesh represents the potato protein-based protofilaments (PoPF) partially coated by orange-coloured filaments connected to each other representing the xanthan based hydrogels (XGH) where the naked part of PoPF (i.e., uncoated by XGH) interacts with the tribo-contact surfaces shown as grey-coloured ball and the rectangular slab. The hydration lubrication mechanism is shown schematically by the transparent water-like spheres attached to the XGH. Zoomed image in the right shows the intact self-assembly outside of the tribo-shear condition.

macroscopic tribology phenomena, where a similar synergistic behaviour of PoPF and XGH was apparent.

Summarising all these complementary suite of techniques, we demonstrate that an electrostatic self-assembly of PoPF and XGH (Fig. 6) provides efficient hydration lubrication fulfilling the high-pressure, low-friction requirements of biological conditions. This unique self-assembly provides robust boundary lubrication via (1) uncovered PoPF anchoring with surfaces effectively; (2) PoPF glueing to the XGH bringing XGH closer to the surface; whilst (3) the exposed, highly hydrated XGH complexed with PoPF providing the hydration lubrication. A key limitation of the study is that conventional, smooth, highly hydrophobic PDMS surfaces were employed in the macrotribology, QCM-D, and molecular dynamics (MD) simulations measurements, whilst negatively charged, hydrophilic (mica) surfaces were used for the atomic force microscopy (AFM) and surface force balance (SFB) experiments. However, it is worth noting that the super-lubricity by these surface-attached protofilament-hydrogel self-assembly can be attributed to the molecular interactions between these boundary layers controlling the frictional dissipation that persisted irrespective of the surfaces used. In addition, the diversity of measurements allowed elucidating the molecular mechanism responsible for such outstanding lubrication properties of this potato protein protofilaments/polysaccharide hydrogel—not only at multiple length scales, but also under varying contact pressure conditions. With these range of measurements and surfaces, we are therefore tackling a wide range of biological applications where pressure many vary from few hundreds to thousands of kilopascals such as oral lubrication to those found in articular joints (MPa), showing the high potential of this aqueous lubricant.

## Conclusions

In summary, we have shown that potato protein, a by-product from starch industry, can be converted into a high functional aqueous lubricant material offering superlubricity. The synergistic association of potato protein-based protofilaments with polysaccharide hydrogels in a patchy architecture leading to the formation of continuous, convoluted filaments, offer unmatched, ultra-low friction coefficient values (of $10^{-2}$ to $10^{-5}$), at moderate-to-extremely high pressures, resembling the remarkable lubrication behaviour of natural biological systems, such as in the oral mucosa, ocular and articular joints[41,45,46]. The superior sustainability of potato proteins further makes them an ideal candidate for the rational design of a next generation of sustainable, hydrogel-based aqueous lubricants. Future studies are focusing on extending such protofilament-hydrogel self-assembly based aqueous lubrication to other plant protein systems besides potato protein and also investigating the lubrication performance in surfaces with various degrees of hydrophobicity.

## Experimental section
### Materials

Potato protein isolate (91% protein content) was purchased from Sosa Ingredients (Barcelona, Spain), citric acid monohydrate (P > 99.5%) from Alfa Aesar (Thermo Fisher Scientific, Lancashire, UK), Decon 90 from Decon Lab Ltd (Hove, UK), ammonia solution (25 wt%) and toluene from Fisher Scientific (Thermo Fisher Scientific Inc, Loughborough, UK), iso-propanol (P99.8%) from MB Fibreglass (Newtownabbey, UK), and xanthan gum (XG), $\kappa$-carrageenan (KC), trisodium citrate dihydrate, hydrochloric acid (HCl, 1 M), sodium azide (NaN$_3$, P > 99.5%), silicon oil, sulfuric acid (P95.0–98.0%), and hydrogen peroxide solution (30 wt%) from Sigma-Aldrich (Gillingham, UK). The SYLGARD™ 184 silicone elastomer kit employed to coat the silicon sensors with polydimethylsiloxane (PDMS) for the quartz-crystal microbalance with dissipation monitoring (QCM-D) experiments was obtained from Dow Chemical Company Ltd (Cheadle, UK), and the silicon monomer and cross-linking agent were mixed at a 10:1 w/w ratio. Both ultrapure water, or Milli-Q grade water (18.2 MΩ cm, Merck Millipore, Bedford, MA, USA), and deuterium oxide (D$_2$O, P99.9%), provided by Sigma-Aldrich (Gillingham, UK), were used in the experiments. Citrate buffer (pH 3.0) was prepared by mixing 10 mM citric acid

monohydrate and 10 mM trisodium citrate dihydrate in adequate proportions so as to reach the appropriate acidic pH. NaN$_3$ (0.02 wt%) was added to all solutions as a preservative. All reagents were used as supplied without any further purification.

### Synthesis of the self-assembled protofilament/hydrogel

Unlike protein-based microgels[16] or microgel-reinforced hydrogels[47,48], protofilaments were first created by acidic hydrolysis of the plant protein, and then assembled with a polysaccharide hydrogel network. The plant protein solution (6.0 wt%) was prepared by adding powdered potato protein isolate in 10 mM citrate buffer at pH 3.0 and stirring for ca. 1.5 h to ensure complete solubilisation. Then, the pH of the solution was adjusted to 3.0 by adding 1 M HCl, the solution was centrifuged at 25,000 × g for 30 min, and finally the supernatant was heated at 65 °C for 30 min to form potato protein protofilaments (PoPF). Xanthan gum (XG)-based hydrogel (XGH, 1.5 wt%) was prepared by dissolving powdered XG in 10 mM citrate buffer at pH 3.0 at 21 ± 2 °C and shearing the solution for 24 h under constant stirring for complete hydration. On the other hand, $\kappa$-carrageenan (KC)-based hydrogel (KCH, 1.5 wt%) was prepared via the dissolution of powdered KC in 10 mM citrate buffer at pH 3.0 by heating at 95 °C while being sheared for 30 min to ensure complete solubilisation. PoPF was added to XGH or KCH dropwise at 21 ± 2 °C, under gentle stirring, to form the self-assembled protofilament/hydrogel at a 2:1 w/w XGH/PoPF or KCH/PoPF ratio, corresponding to a mixture of 1.2 wt% XGH or KCH and 0.6 wt% PoPF.

### Particle size and ζ-potential measurements

Hydrodynamic diameter and surface charge (ζ-potential) measurements were conducted by dynamic light scattering (DLS) on a Zetasizer Ultra instrument (Nano ZS series, Malvern Instruments Ltd, Malvern, UK), at 25 °C, using folded electrophoretic cells (DTS1070, Malvern Instruments Ltd, Malvern, UK) and 100-fold diluted samples.

### Transmission electron microscopy (TEM)

The structure of the self-assembled 2XGH/PoPF protofilament/hydrogel was characterised using TEM, with a FEI Tecnai G2 Spirit-T12 microscope (Thermo Fisher Scientific, Waltham, MA, USA), whose electron gun voltage was fixed at 120 kV. Prior to visualisation, the electron contrast was increased by sample negative staining, using the following protocol: (1) electrostatic cleaning of the homemade, 300 mesh, carbon-coated copper grid with a PELCO easiGlow™ glow discharge cleaning system (Ted Pella Inc, Redding, CA, USA); (2) sample deposition onto the grid, followed by excess liquid blotting and ultrapure water washing; (3) grid staining upon addition of 2.0 wt% uranyl acetate; and (4) air-drying. Nanostructure images were captured using a Gantan CCD camera.

### Small-angle neutron scattering (SANS)

SANS measurements were performed on the SANS2D time-of-flight instrument, at ISIS pulsed neutron source (STFC Rutherford Appleton Laboratory, Didcot, UK)[49], using 1 mm path-length quartz cells (Hellma Analytics, Müllheim, Germany) thermostated at 25 °C with a circulating water bath. SANS2D data were acquired with a polychromatic incident beam of wavelength ($\lambda$) ranging between 1.75 and 16.5 Å, and with a fixed instrument setup of $L1 = L2 = 4$ m (where $L1$ is the collimation length and $L2$ the sample-to-detector distance); a simultaneous scattering vector ($q$)-range of 0.004 to 1 Å$^{-1}$ was achieved using two 1 m$^2$ detectors at 2.4 and 4 m from the sample, with a $q$-resolution varying from ca. 2% at the highest $q$-values to ca. 19% with decreasing $q$-values, calculated using the Mildner Carpenter equation[50]. Detector images were radially averaged and corrected from the scattering of the empty cell and D$_2$O background. Detector efficiency corrections and data normalisation to an appropriate standard were done using MANTID[51].

### Atomic force microscopy (AFM)

The topographic images of the self-assembled 2 XGH/PoPF and 2 KCH/PoPF protofilament/hydrogels, as well as of the individual components

(XGH, KCH, and PoPF), were obtained using an atomic force microscope ((MFP-3D SA, Oxford Instruments Asylum Research, Santa Barbara, CA, USA). A silicon tip on silicon nitride, V-shaped cantilever (length: 115 μm; nominal spring constant: 0.35 N/m) (SNL, Bruker, Camarillo, CA, USA) was used as the probe, in non-contact mode under liquid. Samples were coated onto freshly cleaved mica surfaces by deposition of the dispersion onto the surface and incubation for ca. 30 min, and were then covered with citrate buffer at pH 3.0.

## Molecular dynamics (MD) simulations

The structure of the patatin (main potato protein unit[21]) used in these simulations was found in the protein database (1oxw), and the CHARMM-GUI website was employed to build a model of the resulting PoPF in the MARTINI 3.0.0 forcefield[52]. Using this coarse-grain model, a simulation box was created containing 8 patatin molecules, 66,344 water beads, 720 $Na^+$, and 896 $Cl^-$ ions (which corresponds to 0.15 M of NaCl solution, the quantity of ions required to neutralise the charge of the system). This system was simulated using the protocol prescribed by the CHARMM-GUI website[53], which consists of a minimisation simulation using a soft potential, followed by one using the standard Lennard-Jones potential, and then an equilibration step using the NPT ensemble where a target temperature of 338 K and a target pressure of 1 bar were used. Finally, a production simulation was performed using the NPT ensemble with a target temperature of 338 K and a target pressure of 1 bar. The equilibration simulation was run for 1 ns, and the production NPT simulation for 1 μs. In both simulations, a 20 fs timestep was used. The velocity rescale thermostat was employed in both simulations, while the Berendsen barostat and the Parrinello-Rahman barostat were used in the equilibration simulation and the production simulation, respectively.

The self-assembled PoPF resulting from the coarse-grain simulation was then converted to an all-atom model described by the AMBER forcefield using the CHARMM-GUI[54] implementation of backward.py[55]. This all-atom representation of the system contained 692,689 atoms. This system was then simulated using the simulation protocol for proteins in solution prescribed by CHARMM-GUI[56], which includes a steepest descent energy minimisation simulation, an equilibration simulation using the NVT ensemble in which the Nose-Hoover thermostat is used to control the temperature at 338 K, and then a production simulation using the NPT ensemble with the Nose-Hoover thermostat and the Parrinello Rahman barostat to control the temperature at 338 K and the pressure at 1 bar, respectively. The equilibration simulation was run for 125 ps with a 1 fs timestep, and the production simulation for 100 ns with a 2 fs timestep. The final 50 ns step of this production simulation was used to carry out the analysis of the PoPF in an aqueous solution.

The final configuration from the all-atom simulation of the PoPF in solution was then used as a starting configuration to investigate the interactions of XG molecules with the potato protein nanoparticle. A model of a XG molecule with six repeat units was built using doGlycans[57] and the GLYCAM forcefield[58], which is based on AMBER. Twenty of these model XG molecules were then randomly inserted into the aqueous environment forming XGH around the PoPF, resulting in a system that contains 784,545 atoms. Then the same all-atom simulation protocol was used for this system as was used for the final all-atom simulations used to study the PoPF. However, for this system, the production simulation was run for 200 ns. Again, the final 50 ns step of the production simulation was used to carry out the analysis of the PoPF interacting with the XG molecules in an aqueous solution.

Finally, we used the final configuration from the production simulation of the PoPF interacting with the XGH and placed it such that it was within 1.5 nm of the interface of a slab of polydimethylsiloxane (PDMS). In order to create the slab of PDMS, we first built a single PDMS molecule with 13 monomers using PySoftK[59], and then employed PACKMOL to generate a simulation box with 435 polymers. Using the PolyParGen webserver[60], the AMBER forcefield description of the PDMS polymer was generated. Then, a series of simulations was carried out using the same protocol to generate an amorphous bulk of PDMS as had been previously used for generating an amorphous substrate of another polymer[61]. First, an energy minimisation with a steepest descent algorithm was performed to remove any high-energy steric clashes from the initial configuration. Then the temperature was equilibrated to 800 K in the NVT ensemble using the Berendsen thermostat for 100 ps. Subsequently, the pressure of the simulation box was equilibrated for 5 ns using the Berendsen thermostat and the Berendsen barostat (target pressure of 1 bar). The production simulation was then conducted using the Nosé-Hoover thermostat and the Parrinello-Rahman barostat, to sample from the true NPT ensemble[62,63]. The pressure was kept constant at 1 bar. The simulation began at a temperature of 800 K for 50 ns. Each PDMS molecule was able to diffuse and change its conformation at the high temperature. Following this, a first cooling stage was performed at a constant rate of 10 K ns$^{-1}$ to a temperature of 600 K. Following 50 ns of simulation time, the system was cooled at a constant rate of 10 K ns$^{-1}$ to the final target temperature of 350 K, at which point we observed no structural evolution of the polymer chains. Thus, only a short simulation at the final temperature was required to allow the local conformational changes of the polymer chains to occur. Finally, a slab of PDMS polymers was created by extending the z-dimension in the simulation box and conducting another energy minimisation in order to create two interfaces of the polymer slab.

After inserting PoPF and XGH near the top interface of the PDMS, water molecules and the necessary ions were added to maintain a neutral system and a 150 mM NaCl solution in the simulation box. The simulation box was large enough in the z-dimension such that it would allow the nanoparticle to be more than 3 nm from the other interface of PDMS through the periodic boundary conditions in the z-dimension. This resulted in a system containing a total of 878,345 atoms. We used a similar simulation protocol as was employed for the PoPF and XGH/PoPF simulations. For this simulation, the production simulation was run for 250 ns. Again, the final 50 ns step of the simulation was used for the analysis. In all simulations, the TIP3P water model was employed to describe the water molecules interactions[64].

## Rotational rheometry

The resistance to shearing of the self-assembled 2 XGH/PoPF and 2 KCH/PoPF protofilament/hydrogel, as well as of the individual components (XGH, KCH, PoPF), was assessed with a stress-controlled rheometer (Kinexus ultra + rotational rheometer, Malvern Instruments, Malvern, UK), fitted with a stainless-steel cone/plate geometry (2° angle cone/60 mm diameter (CP2/60) combined with a 65 mm diameter plate (PL65)) and equipped with a temperature-controlled Peltier system (with a ±0.1 °C temperature stability at thermal equilibrium). Apparent shear viscosity ($\eta_{shear}$) was recorded over a shear rate ranging from 0.1 to 1000 s$^{-1}$, at biologically relevant temperature of 37 °C. Each test was repeated at least three times on triplicate samples; the average measurement is shown.

## Extensional rheometry

Resistance to stretching was measured using a HAAKE capillary breakup extensional rheometer (CaBER) 1 (Thermo Electron, Karlsruhe, Germany). The thinning of the midpoint diameter of the capillary bridge generated by the rapid separation of two circular plates (6 mm diameter ($D_o$)) that axially constrained the sample was recorded using a laser micrometre, with a beam thickness of 1 mm and a resolution of 20 μm[47]. The initial separation ($h_o$) between the two circular plates was set at 3 mm, leading to an initial aspect ratio ($h_0/D_0$) of 0.5. The final axial displacement ($h_f$) was set at 10 mm in 50 ms to allow filament thinning. Each sample (ca. 0.1 mL) was injected between the plates using a 1 mL syringe. The experiment was triggered 60 s after loading the sample, to limit shear and temperature preconditioning effects. At least five repetitions were performed at 37 °C. High-speed videos of the experiments were also taken at 1000 frames/s, using a PhantomV1612 high-speed camera (Vision Research, Wayne, NJ, USA), to record the shape evolution of the capillary thread. Due to the displacement of the midpoint of the filament, the images acquired were processed using the ImageJ software to detect the filament interface, and compared to the data acquired with the

laser micrometre. For an upper convected Maxwell model, the elastocapillary force balance predicts an exponential diameter decay in time with a characteristic relaxation time ($\lambda_c$)[65]:

$$D_{\min}(t) \sim e^{-\frac{t}{3\lambda_c}} \tag{1}$$

where $D_{\min}$ is the instantaneous filament midpoint diameter. Surface tension measurements were performed in triplicates using the Wilhelmy plate method (Kruss ST10, KRÜSS GmbH, Hamburg, Germany) (Supplementary Fig. 13), at 37 °C and minimum speed (0.5 mm min$^{-1}$) to limit the influence of the shear generated between the sample and measuring plate; the average measurement is shown.

The cylindrical elements of the self-assembled 2 XGH/PoPF and 2 KCH/PoPF protofilament/hydrogel, as well as of the individual components (XGH, KCH, PoPF), at the axial mid-plane plate were subject to a strain ($\varepsilon$) expressed as:

$$\varepsilon = -2 \ln \frac{D_{\min}}{D_o} \tag{2}$$

The instantaneous strain rate ($\dot{\varepsilon}$) for a cylindrical element of fluid is given by:

$$\dot{\varepsilon} = \frac{-2}{D_{\min}} \frac{dD_{\min}}{dt} \tag{3}$$

The apparent extensional viscosity of the liquid ($\eta_{extensional}$) is therefore expressed as:

$$\eta_{extensional} = \frac{2\sigma/D_{\min}}{\dot{\varepsilon}} \tag{4}$$

For coherence with a recent study[47,48], the transient Trouton ratio ($T_r$) was computed as the ratio between the apparent extensional and shear viscosity:

$$T_r(\dot{\varepsilon}) = \frac{\eta_{extensional}(\dot{\varepsilon})}{\eta_{shear}(\dot{\varepsilon})} \tag{5}$$

where the dependence of shear rate on the strain rate has been considered at the denominator.

## Macroscale tribology

The lubrication performance was evaluated by tribology, using a conventional mini-traction machine (MTM2, PCS Instruments, London, UK) in combination with smooth hydrophobic elastomeric surfaces, i.e., a PDMS ball (19 mm diameter) and disc (46 mm diameter) in a sliding/rolling motion, displaying a 50 nm surface roughness and 2.0 MPa Young's modulus[47,66]. A constant normal force of 2.0 N, corresponding to a Hertzian contact pressure of ca. 300 kPa[35], and a temperature of 37 °C were applied. The relative motion of the rolling and sliding surfaces is represented by the entrainment speed, which is the average of the ball and disc linear speeds at the contact point, the sliding/rolling ratio being fixed at 50% (i.e., the contribution of both rolling and sliding to motion being defined as equal). The evolution of the friction coefficient was recorded over an entrainment speed range of 0.0035–1.0 m s$^{-1}$. The thickness ($h_{\min}$) of the hydrodynamic fluid-film for the samples was estimated using the following equation[35,67–69]:

$$h_{\min} = 2.8R'U^{0.68}W^{-0.20} \tag{6}$$

where $U$ is the dimensionless speed parameter ($\frac{u\eta_\infty}{E'r'}$), $W$ is the dimensionless load parameter ($\frac{F_N}{E'r'^2}$), $\eta_\infty$ is the viscosity ($\eta$) of the fluid-film at the tribologically relevant high-shear rates ($\dot{\gamma}$) at 1000 s$^{-1}$ and $u$ is the entrainment speed. The $\eta_\infty$ often is taken as the limiting high-shear viscosity of the fluid in the rheological measurements (the second plateau in the $\eta$–$\dot{\gamma}$

graphs) and is a measure of the hydrodynamic forces generated by the fluid-film during tribo-contacts[67,68]. Each experiment was repeated at least three times on triplicate samples; the average measurement is shown.

## Surface force balance (SFB) measurements

The SFB setup (Supplementary Fig. 14) and the measurement procedures have been described previously in detail[37,38]. Briefly, the cleaved molecularly smooth mica sheets (each ca. 2.5 μm thick) were back-silvered and then glued on hemicylindrical quartz lenses of radius of 10 mm, which were oppositely mounted in a crossed-cylinder orientation in the SFB apparatus. The closest separation between two mica surfaces ($D$) was measured from the wavelength of fringes of equal chromatic order (FECO) (accuracy ca. 2–3 Å).

The normal forces ($F_n$) were determined from the bending ($\Delta D_0 - \Delta D$) of the horizontal spring, as follows:

$$F_n(D) = F_n(D - \Delta D) + K_n(\Delta D_0 - \Delta D) \tag{7}$$

where $\Delta D_0$ is the applied normal motion, $\Delta D$ the change of surface separation, and $K_n$ the horizontal (normal force) spring constant.

The shear forces ($F_s$) were determined through measuring the bending $\Delta X$ of the vertical spring, as follows:

$$F_s = K_s \Delta X \tag{8}$$

where $K_s$ is the vertical (shear force) spring constant.

We estimate an uncertainty of $\pm 10^{-5}$ in the SFB measurements arising from thermal drift and optical fringe errors[41].

## Quartz-crystal microbalance with dissipation monitoring (QCM-D) measurements

The capacity to adsorb onto a hydrophobic surface and to remain attached following rinsing was evaluated by using a QCM-D (Q-Sense E4 system, Biolin Scientific AB, Västra Frölunda, Sweden), equipped with PDMS-coated sensors[4].

Silica-coated QCM-D sensors (QSX-303, Q-Sense, Biolin Scientific AB, Västra Frölunda, Sweden) were first treated by UV/Ozone for 15 min to generate hydrophilic surfaces, and then immersed into sulfuric acid for 1 h, before being sonicated twice in ultrapure water for 10 min and dried under nitrogen. The substrates were further cleaned by immersing them into an RCA silicon wafer cleaning solution (made up of 5:1:1 ultrapure water/ammonia/30% hydrogen peroxide) at 80 °C, for 15 min, to remove any remaining organic/insoluble impurities, and by subjecting them to three cycles of 10-min sonication in ultrapure water, before drying them again under nitrogen. Cleaned surfaces were spin-coated with PDMS (prepared in toluene at a concentration 0.5 wt%) at 5000 RPM (with a 2500 RPM/s acceleration), for 60 s, and finally placed under vacuum overnight, at 80 °C, to ensure efficient PDMS cross-linking.

Prior to any measurement, PDMS-coated silicon substrates were thoroughly cleaned through sequential immersion in toluene (30 s), iso-propanol (30 s), and ultrapure water (5 min), before rinsing them extensively with ultrapure water, and drying them under nitrogen. Once assembled, the QCM-D flow cells were prefilled with citrate buffer until reaching a stable baseline. Each sample was diluted (i.e., 20-fold dilution in citrate buffer) before being injected into the PDMS-coated silica sensor-containing cell. Once frequency and dissipation reached a plateau, characteristic of adsorption saturation, buffer was flushed into the cell, to study the desorption behaviour of the surface-adsorbed lubricant. Solutions were injected at a flow rate of 100 μL/min, and measurements conducted at a temperature of 25 ± 2 °C. Changes in resonance frequency ($\Delta f$) were recorded simultaneously over time. Each experiment was reproduced at least three times; a representative curve is shown for the 5th overtone.

## Data availability

The raw data in support of most of the quantitative figures reported in this work are available from the corresponding author upon reasonable request.

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

## Acknowledgements

The authors acknowledge the European Research Council (ERC) for the provision of funding under the European Union's Horizon 2020 research and innovation programme (grant agreements: no. 890644) and UKRI Horizon Europe Guarantee Fund (grant agreement: no. EP/X03514X/1). The authors also thank the UKRI Healthy Ageing Catalyst Award (ES/X006565/1) for supporting this research. O.P. acknowledges: the British Society of Rheology (BSR) for the award of the Undergraduate Summer Research Bursary, and the Royal Society for the award of the International Exchanges Scheme that allowed her to conduct a research visit at Weizmann Institute. O.P. thanks ISIS for the provision of beam time on SANS2D (https://doi.org/10.5286/ISIS.E.RB2000279). This work benefitted from the use of the SasView application, originally developed under NSF award DMR-0520547. SasView contains code developed with funding from the European Union's Horizon 2020 research and innovation programme under the SINE2020 project, grant agreement No. 654000.

## Author contributions

O.P., J.K., C.L., M.R., and A.S. conceptualised the study and designed the experimental protocol. A.S. supervised the study. O.P. designed the study protocol. O.P., M.M., Y.M., and E.L. performed the rotational rheology, macroscale tribology, and QCM-D experiments. O.P. conducted the TEM and AFM measurements with the guidance of N.K. and J.K. O.P. thanks Dr Nadav Elad for his technical support with the TEM experiments. O.P. performed the SANS measurements with the help of J.D. and A.T. C.L. designed and carried out the MD simulation study. Y.D., N.K., and J.K. designed the SFB experiments and performed the data analyses. A.A.S. and M.R. developed and conducted the extensional rheology experiments. O.P., Y.D., A.A.S., C.L., and A.S. prepared the figures, and wrote the main manuscript text. All authors reviewed the manuscript.

## Competing interests

The authors declare no competing interests.
