## [Peer Review File · Communications Materials]

26th May 24

Dear Professor Sarkar,

Thank you for submitting your manuscript, "Self-assembly of sustainable plant protein protofilaments into a hydrogel for ultra-low friction across length scales", to Communications Materials. It has now been seen by 2 referees, whose comments are appended below. You will see that while they find your work of interest, some important points are raised. We are interested in the possibility of publishing your study in Communications Materials, but would like to consider your response to these concerns in the form of a revised manuscript before we make a decision on publication.

In particular, Reviewer 1 mentions that the current conclusions are weak and require further mechanism explanation, as well as asking for an analysis against other materials. In addition, Reviewer 2 asks why the potato proteins need to be converted into amyloid fibrils and not use other protein aggregates.

We therefore invite you to revise and resubmit your manuscript, taking into account the points raised.

When submitting your revised manuscript, please include the following:

-A response letter with a point-by-point reply to each of the referee comments and a description of changes made. Please include the complete referee report in the response letter. Please note that the response letter must be separate to the cover letter to the editors.

-A marked-up version of the manuscript with all changes to the text in a different colored font. Please do not include tracked changes or comments. Please select the file type 'Revised Manuscript - Marked Up' when uploading the manuscript file to our online system.

-A clean version of the manuscript. Please select the file type 'Article File'.

-An updated Editorial Policy checklist, uploaded as a 'Related Manuscript File' type. This checklist is to ensure your paper complies with all relevant editorial policies. If needed, please revise your manuscript in response to these points. Please note that this form is a dynamic 'smart pdf' and must therefore be downloaded and completed in Adobe Reader. Clicking this link will download a zip file containing the pdf.

In addition, please ensure that the following requirements are met, and that any other relevant checklists are completed and uploaded under the 'Related Manuscript file' type with the revised article.

Chemical and biomolecular materials: Characterization of chemical and biomolecular materials

In the event that your manuscript is accepted we will provide detailed guidance on our journal policies and formatting. You may however wish to ensure that the manuscript complies with our house style at this stage. See our style and formatting guide (<https://www.nature.com/documents/commsj-phys-style-formatting-guide-accept.pdf>) and checklist (<https://www.nature.com/documents/commsj-phys-style-formatting-checklist-article.pdf>) for reference.

Data availability statements and data citations policy: All Communications Materials manuscripts must include a section titled "Data Availability" at the end of the Methods section or main text (if no Methods). More information on this policy, and a list of examples, is available at <http://www.nature.com/authors/policies/data/data-availability-statements-data-citations.pdf>.

- Accession codes for deposited data
- Other unique identifiers (such as DOIs and hyperlinks for any other datasets)
- At a minimum, a statement confirming that all relevant data are available from the authors
- If applicable, a statement regarding data available with restrictions
- If a dataset has a Digital Object Identifier (DOI) as its unique identifier, we strongly encourage including this in the Reference list and citing the dataset in the Data Availability Statement.

DATA SOURCES: We strongly encourage authors to deposit all new data associated with the paper in a persistent repository where they can be freely and enduringly accessed. We recommend submitting the data to discipline-specific, community-recognized repositories, where possible and

a list of recommended repositories is provided at <http://www.nature.com/sdata/policies/repositories>.

If a community resource is unavailable, data can be submitted to generalist repositories such as figshare or Dryad Digital Repository. Please provide a unique identifier for the data (for example a DOI or a permanent URL) in the data availability statement, if possible. If the repository does not provide identifiers, we encourage authors to supply the search terms that will return the data. For data that have been obtained from publically available sources, please provide a URL and the specific data product name in the data availability statement. Data with a DOI should be further cited in the methods reference section.

Please use the following link to submit your documents:

[link readcted]

We hope to receive your revised paper within three months; please let us know if you aren't able to submit it within this time so that we can discuss how best to proceed. If we don't hear from you, and the revision process takes significantly longer, we will close your file. In this event, we will still be happy to reconsider your paper at a later date, as long as nothing similar has been accepted for publication at Communications Materials or published elsewhere in the meantime.

Please do not hesitate to contact me if you have any questions or would like to discuss these revisions further. We look forward to seeing the revised manuscript and thank you for the opportunity to review your work.

Best regards,

Dr Jet-Sing Lee

Associate Editor

Communications Materials

orcid.org/0000-0002-6740-8700

Reviewers' comments:

Reviewer #1 (Remarks to the Author):

This article presents an extensive examination of the multiscale structure-property relationships of plant protein filaments and their self-assembly within polysaccharide hydrogels (composed of either xanthan gum or k-carrageenan). The authors deploy a veritable panoply of characterisation tools to investigate the self-assembly and lubricity of their materials and constituents from the atomic scale through the macroscale: molecular dynamics simulation, atomic force microscopy, transmission electron microscopy, neutron scattering, surfaces forces balance, quartz crystal microbalance, dynamic light scattering, rheology, and mini-traction machines. The manuscript would be considered fit for publication in this journal were the following questions and concerns addressed:

1. The authors are commended for their efforts to apply so many diverse techniques in their investigations. However, the manuscript would greatly benefit from greater attention to detail in the presentation, motivation, and interpretation of the measurements to avoid the impression of taking everything but the proverbial kitchen sink. For example, the molecular dynamics simulations, QCM, and macrotribology measurements involve interactions with PDMS, yet SFB measurements do not.
2. Special care should be taken to avoid the impression that friction (or superlubricity, which is very low friction indeed) is a material property, as is stated in the first sentence of the abstract.
3. The unprecedented friction coefficients of 0.004 to 0.00007 are indeed very low, but it would be useful to indicate that these measurements were the result of different experimental techniques. The macroscale tribology measurement reports “friction coefficient values as low as 0.004 +/- 0.005” but it appears as though the standard deviation is as large or higher than the reported value. What is the measurement uncertainty of the friction coefficient in this study? What is the lowest measurement of friction coefficient possible with this experimental configuration?
4. What is the estimated fluid film thickness during macroscale and nanoscale tribology measurements from soft elasto-hydrodynamic lubrication theory? Is it possible that the contact configuration and conditions fall outside of the classical boundary lubrication regime?

5. The authors propose a new material that is quite complicated to consider, especially across multiple scales, without the benefit of illustrations or schematics. It would be useful to have a schematic with dimensions of protofilaments, how these protein-based protofilaments self-assemble both outside and within polysaccharide hydrogels.

6. The conclusions, following such extensive examination, are rather weak. The mechanism of low friction could benefit from a deeper analysis of the hydration lubrication mechanism. How many hydration layers are present on the hydrogel polysaccharide, and how many on the protofilament? What could only have been known through the wealth of data collected here that would have otherwise never have been understood with fewer available techniques? It would be helpful for the authors to comment on the similarities and differences between the instruments and characterisation tools employed; which were strictly necessary? Which overlapped or otherwise agreed with other techniques? Which gaps remain, and at which length-scales, time-scales, force-scales? Are these knowledge gaps still essential for the complete understanding of the material systems in this manuscript? If so, what is the plan for future investigations? What obvious experimental techniques were not used and why not? For instance, the authors used AFM but not lateral force microscopy to measure friction at the nanoscale, instead opting for SFB.

7. The manuscript would benefit from a comparative analysis of the materials against other well-known low-friction material systems. Which mechanisms do the materials in the manuscript share with these?

Thank you for the opportunity to review this manuscript.

Reviewer #2 (Remarks to the Author):

This manuscript reports that the assemblies formed from protein nanofibrils and xanthan gum have good lubricating properties and have the great potential to create oil-free foods. Overall, the data in the manuscript are comprehensive and corroborated with each other. There are two points that need to be taken into consideration:

1. Why is it necessary to convert potato proteins into amyloid fibrils and not use other protein aggregates? Is it because the fibrils require a lower gelling concentration or something else.
2. The article is a bit wordy and lengthy and needs to be simplified.

8Reviewer #1:

This article presents an extensive examination of the multiscale structure-property relationships of plant protein filaments and their self-assembly within polysaccharide hydrogels (composed of either xanthan gum or k-carrageenan). The authors deploy a veritable panoply of characterisation tools to investigate the self-assembly and lubricity of their materials and constituents from the atomic scale through the macroscale: molecular dynamics simulation, atomic force microscopy, transmission electron microscopy, neutron scattering, surfaces forces balance, quartz crystal microbalance, dynamic light scattering, rheology, and mini-traction machines. The manuscript would be considered fit for publication in this journal were the following questions and concerns addressed:

****Response:** We thank the reviewer for highlighting the multiscale importance of this study and also the suite of complementary techniques used. We have addressed the comments highlighted by the reviewer and revised the manuscript accordingly.

1. The authors are commended for their efforts to apply so many diverse techniques in their investigations. However, the manuscript would greatly benefit from greater attention to detail in the presentation, motivation, and interpretation of the measurements to avoid the impression of taking everything but the proverbial kitchen sink. For example, the molecular dynamics simulations, QCM, and macrotribology measurements involve interactions with PDMS, yet SFB measurements do not.

****Response:** Indeed, Reviewer 1 is correct when stating that different surfaces were used to conduct our tribology, structural, and interfacial studies. The idea behind such a diversity of measurements was to elucidate the molecular mechanism responsible for such outstanding lubrication properties of the newly developed plant-based potato protein protofilaments/polysaccharide hydrogel – not only at multiple length scales, but also under varying contact pressure conditions. With these range of measurements and surfaces, we are therefore tackling a wide range of biological applications where pressure many vary from hundreds to few thousands of kilopascals such as oral lubrication to those found in articular joints, showing the high potential of this aqueous lubricant. Additionally, in all cases, we are dealing mainly with boundary lubrication by the surface-attached boundary layers, where the molecular interactions between these boundary layers control the frictional dissipation and such ultralow friction persists independent of the surfaces used. Nevertheless, we have added a limitation highlighting this.

Lines 554-570 in the revised manuscript now read as “A key limitation of the study is that conventional, smooth, highly hydrophobic PDMS surfaces were employed in the macrotribology, quartz-crystal microbalance with dissipation monitoring (QCM-D), and molecular dynamics (MD) simulations measurements, whilst negatively charged, hydrophilic (mica) surfaces were used for the atomic force microscopy (AFM) and surface force balance (SFB) experiments. However, it is worth noting that the superlubricity by these surface-attached protofilament-hydrogel self-assembly can be attributed to the molecular interactions between these boundary layers controlling the frictional dissipation that persisted irrespective of the surfaces used. In addition, the diversity of measurements allowed elucidating the molecular mechanism responsible for such outstanding lubrication properties of the newly developed plant-based potato protein protofilaments/polysaccharide hydrogel – not only at

multiple length scales, but also under varying contact pressure conditions. With these range of measurements and surfaces, we are therefore tackling a wide range of biological applications where pressure many vary from few hundreds to thousands of kilopascals such as oral lubrication to those found in articular joints, showing the high potential of this aqueous lubricant.”

2. Special care should be taken to avoid the impression that friction (or superlubricity, which is very low friction indeed) is a material property, as is stated in the first sentence of the abstract.

****Response:** We agree the phrasing was not correct. We have changed this to “performance indicator” rather than material property in **Lines 35-36** in the revised manuscript.

3. The unprecedented friction coefficients of 0.004 to 0.00007 are indeed very low, but it would be useful to indicate that these measurements were the result of different experimental techniques. The macroscale tribology measurement reports “friction coefficient values as low as 0.004 +/- 0.005” but it appears as though the standard deviation is as large or higher than the reported value. What is the measurement uncertainty of the friction coefficient in this study? What is the lowest measurement of friction coefficient possible with this experimental configuration?

****Response:** The friction coefficient value of $7 \cdot 10^{-5}$ was obtained with the SFB; the experimental uncertainty associated with it has now been included in the revised version of the manuscript in **Lines 854-855** in the revised manuscript, which now read as: We estimate an uncertainty of $\pm 10^{-5}$ in the SFB measurements arising from thermal drift and optical fringe errors⁴¹.

4. What is the estimated fluid film thickness during macroscale and nanoscale tribology measurements from soft elasto-hydrodynamic lubrication theory? Is it possible that the contact configuration and conditions fall outside of the classical boundary lubrication regime?

****Response:** Irrespective of the scales, we are in the boundary regime. In agreement with the reviewer, we have calculated the soft elasto-hydrodynamic film thickness in the macrotribology and plotted the data in **Figure S8, Lines 822-832** in the revised manuscript in the method section read as “The thickness (h_{min}) of the hydrodynamic fluid-film for the samples was estimated using the following equation ^{35, 67, 68, 69}:

$$h_{min} = 2.8R'U^{0.68}W^{-0.20} \quad (6)$$

where, U is the dimensionless speed parameter ($\frac{u\eta_{\infty}}{E'r'^2}$), W is the dimensionless load parameter ($\frac{FN}{E'r'^2}$), η_{∞} is the viscosity (η) of the fluid-film at the tribologically relevant high shear rates ($\dot{\gamma}$) at 1000 s^{-1} and u is the entrainment speed. The η_{∞} often is taken as the limiting high-shear viscosity of the fluid in the rheological measurements (the second plateau in the $\eta - \dot{\gamma}$ graphs) and is a measure of the hydrodynamic forces generated by the fluid-film during tribo-contacts ^{67, 68}. “

Lines **353-358** in the result section read as “When plotted against minimum hydrodynamic film thickness (h_{min}) (**Figure S8**), the friction curves of the constituents overlapped with that of

the buffer. On the contrary, the self-assembly (2 XGH/PoPF system) showed ultra-low friction independent of the film thickness, before reaching hydrodynamic regime, showing effective boundary films capable of maintaining low friction forces at low surface separations.”

On the other hand, scenario as we usually see it in the SFB is that the two coated mica surfaces compress against each other till the load is balanced by the steric forces due to the boundary layer (xanthan gum/potato protein in this case), so here we are probing at molecular scale and there is no conditions where we are out of boundary regime. The high friction conditions are then due to molecular interactions (*e.g.*, bond breaking/reforming) as the boundary layers slide past each other, while the low friction arises from the shear of the sub-nanometer hydration layers in the hydration lubrication mechanism, together with passage over energy barriers as described for example in Reference 36.

We have now added a text in **Lines 534 to 539** in the revised manuscript which read as: “The high friction conditions are then largely due to molecular interactions (*e.g.* bond breaking/reforming) as the boundary layers slide past each other, while the low friction arises from much-reduced dissipation due to shear of the sub-nanometer hydration layers in the hydration lubrication mechanism, together with passage over energy barriers as described previously³⁶ (where phonons are generated leading to the weak dissipation observed).”

5. The authors propose a new material that is quite complicated to consider, especially across multiple scales, without the benefit of illustrations or schematics. It would be useful to have a schematic with dimensions of protofilaments, how these protein-based protofilaments self-assemble both outside and within polysaccharide hydrogels.

****Response:** Although we have already described the mechanism via the MD simulation, we agree with the idea of Reviewer 1 and have added a schematic illustration in **Figure 6** describing the overall mechanism. Lines **548-554** in the revised manuscript read as “Summarising all these complementary suite of techniques, we demonstrate that an electrostatic self-assembly of PoPF and XGH (**Figure 6**) provides efficient hydration lubrication fulfilling the high-pressure, low-friction requirements of biological conditions. This unique self-assembly provides a robust boundary lubrication via 1) uncovered PoPF anchoring with surfaces effectively; 2) PoPF glueing to the XGH bringing XGH closer to the surface; whilst 3) the exposed, highly hydrated XGH complexed with PoPF providing the hydration lubrication.”

6. The conclusions, following such extensive examination, are rather weak. The mechanism of low friction could benefit from a deeper analysis of the hydration lubrication mechanism. How many hydration layers are present on the hydrogel polysaccharide, and how many on the protofilament? What could only have been known through the wealth of data collected here that would have otherwise never have been understood with fewer available techniques? It would be helpful for the authors to comment on the similarities and differences between the instruments and characterisation tools employed; which were strictly necessary? Which overlapped or otherwise agreed with other techniques? Which gaps remain, and at which length-scales, time-scales, force-scales? Are these knowledge gaps still essential for the complete understanding of the material systems in this manuscript? If so, what is the plan for future investigations? What obvious experimental techniques were not used and why not?

For instance, the authors used AFM but not lateral force microscopy to measure friction at the nanoscale, instead opting for SFB.

****Response:** As highlighted in the response to the Point no. 5, we have now also included a schematic (**Figure 6**) and have a very clear picture that tell us about the synergy of the XGH and PoPF boundary layers in acting as a lubricating boundary layer, which is by no means *a priori* obvious and we have added this in the text.

Lines **548-554** in the revised manuscript read as “Summarising all these complementary suite of techniques, we demonstrate that an electrostatic self-assembly of PoPF and XGH (**Figure 6**) provides efficient hydration lubrication fulfilling the high-pressure, low-friction requirements of biological conditions. This unique self-assembly provides a robust boundary lubrication via 1) uncovered PoPF anchoring with surfaces effectively; 2) PoPF glueing to the XGH bringing XGH closer to the surface; whilst 3) the exposed, highly hydrated XGH complexed with PoPF providing the hydration lubrication.”

It is challenging to know precisely how many hydration layers are present on the XGH, and even if we did, we do not have a good quantitative model for relating number of hydration layers with the friction coefficient. What is important is that there is hydration at the slip-plane between the layers, which acts through the hydration lubrication mechanism to lower the friction. Lateral force microscopy is seriously problematic for such soft layers, as the AFM tip tends to ‘plough’ through the soft boundary layers, distorting the true friction as between two affine surfaces. Certainly the SFB data unambiguously identifies the synergy of the two main components in creating a hydrated low friction boundary layer, and indicates the extent of surface attachment. We agree with the reviewer, we should add future plans. So we have included text for future work in the conclusion.

Lines 591-594 now state that “Future studies are focusing on extending such protofilament-hydrogel self-assembly based aqueous lubrication to other plant protein systems besides potato protein and also investigating the lubrication performance in surfaces with various degrees of hydrophobicity.”

7. The manuscript would benefit from a comparative analysis of the materials against other well-known low-friction material systems. Which mechanisms do the materials in the manuscript share with these? Thank you for the opportunity to review this manuscript.

****Response:** Phosphatidylcholine (PC) lipids are comparable strongly lubricating boundary materials, where the strong headgroup hydration in some sense plays the role of the hydrated xanthan gum monomers in enabling hydration lubrication. The related Reference **41** is already included in the reference and discussed in the manuscript.

Reviewer #2:

This manuscript reports that the assemblies formed from protein nanofibrils and xanthan gum have good lubricating properties and have the great potential to create oil-free foods. Overall, the data in the manuscript are comprehensive and corroborated with each other. There are two points that need to be taken into consideration:

****Response:** We thank the reviewer for highlighting the comprehensiveness of the manuscript and also echoing the application. We have addressed the two comments highlighted by the reviewer.

1. Why is it necessary to convert potato proteins into amyloid fibrils and not use other protein aggregates? Is it because the fibrils require a lower gelling concentration or something else.

****Response:** Generally, plant proteins tend to aggregate due to their surface hydrophobicities. Such aggregates produce very high friction due to jamming of the contact as shown in our previous work (*Food Hydrocolloids*, Volume 116, 2021, 106636; *Nature Communications*, Volume 14, 2023, 4743). The protofilaments partially coated with hydrated biopolymers forming hydrogel offer easy entrainment by virtue of length scale that tend to align in the tribocontact and form the boundary layers offering hydration lubrication. We have added some suggestions for future work. As highlighted in revised text in conclusions in **Lines 591-594** “Future studies are focusing on extending such protofilament-hydrogel self-assembly based aqueous lubrication to other plant protein systems besides potato protein and also investigating the lubrication performance in surfaces with various degrees of hydrophobicity.”, we are now evaluating if such self-assembly technology of protofilament and hydrogel can be extended to other plant proteins.

2. The article is a bit wordy and lengthy and needs to be simplified.

****Response:** We thank the reviewer for this suggestion and have reduced the length and clarified sentences where possible without comprising the scientific discussion in the manuscript.

11th Jul 24

Dear Professor Sarkar,

Your manuscript titled "Self-assembly of sustainable plant protein protofilaments into a hydrogel for ultra-low friction across length scales" has now been seen again by our referees, whose comments appear below. In light of their advice I am delighted to say that we are happy, in principle, to publish a suitably revised version in Communications Materials under the open access CC BY license (Creative Commons Attribution v4.0 International License).

We therefore invite you to edit your manuscript to comply with our journal policies and formatting style in order to maximise the accessibility and therefore the impact of your work.

EDITORIAL REQUESTS

* Your manuscript should comply with our policies and format requirements, detailed in our style and formatting guide (<https://www.nature.com/documents/commsj-phys-style-formatting-guide-accept.pdf>).

* Please edit your manuscript according to the editorial requests in the attached table, and outline revisions made in the right hand column. If you have any questions or concerns about any of our requests, please do not hesitate to contact me. It is important that each request be addressed in order to avoid delays in accepting your manuscript. Please upload the completed table with your manuscript files as a Related Manuscript file.

* The editorial requests table also includes a full list of the files that must be provided upon resubmission. Please upload your files according to this table.

* An updated editorial policy checklist that verifies compliance with all required editorial policies must be completed and uploaded with the revised manuscript. All points on the policy checklist must be addressed; if needed, please revise your manuscript in response to these points. Please note that this form is a dynamic 'smart pdf' and must therefore be downloaded and completed in Adobe Reader. Clicking this link will download a zip file containing the pdf.

OPEN ACCESS

Communications Materials is a fully open access journal. Articles are made freely accessible on publication under a CC BY license (Creative Commons Attribution 4.0 International License). This license allows maximum dissemination and re-use of open access materials and is preferred by many research funding bodies.

For further information about article processing charges, open access funding, and advice and support from Nature Research, please visit <https://www.nature.com/commsmat/about/open-access>

RESUBMISSION

At acceptance, you will be provided with instructions for completing this CC BY license on behalf of all authors. This grants us the necessary permissions to publish your paper. Additionally, you will be asked to declare that all required third party permissions have been obtained, and to provide billing information in order to pay the article-processing charge (APC).

Please use the following link to submit your revised files:

[link redacted]

We hope to hear from you within two weeks; please let us know if the process may take longer.

Best regards,

Dr Jet-Sing Lee

Associate Editor

Communications Materials

orcid.org/0000-0002-6740-8700

REVIEWERS' COMMENTS:

Reviewer #1 (Remarks to the Author):

The authors have made acceptable corrections and changes to the manuscript and it is acceptable for publication.

Reviewer #2 (Remarks to the Author):

the revised manuscript can be published.